# Laser Grinding of Single-Crystal Silicon Wafer for Surface Finishing and Electrical Properties

**DOI:** 10.3390/mi12030262

**Published:** 2021-03-04

**Authors:** Xinxin Li, Yimeng Wang, Yingchun Guan

**Affiliations:** 1School of Mechanical Engineering and Automation, Beihang University, 37 Xueyuan Road, Beijing 100083, China; lixx0912@buaa.edu.cn (X.L.); ymwang1007@buaa.edu.cn (Y.W.); 2National Engineering Laboratory of Additive Manufacturing for Large Metallic Components, Beihang University, 37 Xueyuan Road, Beijing 100083, China; 3International Research Institute for Multidisciplinary Science, Beihang University, 37 Xueyuan Road, Beijing 100083, China; 4Ningbo Innovation Research Institute, Beihang University, Beilun District, Ningbo 315800, China

**Keywords:** laser grinding, nanosecond laser, single-crystal silicon wafer, surface finishing, resistivity

## Abstract

In this paper, we first report the laser grinding method for a single-crystal silicon wafer machined by diamond sawing. 3D laser scanning confocal microscope (LSCM), X-ray diffraction (XRD), scanning electron microscope (SEM), X-ray photoelectron spectroscopy (XPS), laser micro-Raman spectroscopy were utilized to characterize the surface quality of laser-grinded Si. Results show that SiO_2_ layer derived from mechanical machining process has been efficiently removed after laser grinding. Surface roughness Ra has been reduced from original 400 nm to 75 nm. No obvious damages such as micro-cracks or micro-holes have been observed at the laser-grinded surface. In addition, laser grinding causes little effect on the resistivity of single-crystal silicon wafer. The insights obtained in this study provide a facile method for laser grinding silicon wafer to realize highly efficient grinding on demand.

## 1. Introduction

Single-crystal silicon wafer has been widely used in semiconductor applications including computer systems, telecommunications equipment, automobiles, consumer electronics, automation and control systems, analytical and defense systems [1]. High-quality silicon wafers without damages are essential for these applications. In semiconductor industry, silicon wafer is usually produced by slicing, edge profiling, lapping, grinding, etching, grinding, and cleaning processes [2,3]. Damages such as amorphous layers, dislocations, and microcracks, can be produced at the surface of silicon wafers during mechanical machining processes [4,5]. These damages will reduce performance and lifetime of the wafers [6]. Although the post processes such as chemo-mechanical polishing (CMP) process [7] and etched-wafer fine grinding [8] can reduce damaged layers, it increases the total costs significantly.

Laser surface microprocessing has been an emerging technology, which has many advantages rather than traditional methods including non-contact, environmentally friendly, and high flexibility [9,10]. Till now, laser microprocessing such as laser recovery, laser microtexturing, laser annealing, and laser drilling have been successfully used for semiconductor materials. Yan et al. reported that the amorphous layers transformed to single-crystal silicon and dislocations and microcracks were completely eliminated by using nanosecond-pulsed laser. Moreover, the roughness of the surface was reduced from RMS = 12 nm to 8 nm after laser recovery [4,11,12,13]. Gupta et al. have used nanosecond pulse laser annealing to successfully demonstrate the phase evolution of single-crystal silicon wafer with different pulse widths and laser fluences [14]. Mahdieh et al. have reported that the roughness of amorphous silicon wafer was reduced from RMS 9 nm to 0.5 nm after nanosecond laser annealing [15]. Wang et al. successful obtained nanostructure on amorphous thin silicon film surface using femtosecond pulse laser and found the absorptance of Si thin film increased after laser irradiation [16]. Zheng et al. produced regular arrays of sub-micron bumps on single-crystal silicon wafer by using a continuous wave fiber laser [17]. Jiao et al. generated holes on single-crystal silicon wafer using femtosecond laser and the spatter area decreased and drilling efficiency increased with substrate temperature increase [18]. However, there is little discussion in the literature using laser grinding method to improve the surface quality of single-crystal silicon wafer.

The objective of this study is to investigate the effect of nanosecond pulsed laser grinding on surface improvement of single-crystal silicon wafer. The influence of laser grinding on the surface modification, microstructure, and resistivity of single-crystal silicon wafer was studied. Roughness reduction of single-crystal silicon wafer before and after laser grinding were evaluated. Moreover, the evolution of chemical composition, crystallinity and resistivity of single-crystal silicon wafer surface were discussed during laser grinding.

## 2. Materials and Methods

Single-crystal silicon wafer machined by diamond sawing were used, with the thickness of 525 μm, orientation of <100>, with the boron doping concentration of 10^16^/cm^3^, with the diameter of 4 inch.

The 1064 nm wavelength nanosecond pulse laser with a full laser power of 100 W at the 100% set point and a laser pulse width of 220 ns is employed to irradiated single-crystal silicon wafer. Figure 1 illustrates the schematic for the experimental setup of laser grinding silicon wafer. In order to focus and scan the laser beam in the horizontal and vertical directions, a two-mirror galvanometric scanner with an F-theta objective lens is used. The focal beam diameter of 35 μm can be achieved and the beam has a Gaussian energy distribution. Formation of typical morphologies and microstructure is the result of synergistic combination between laser power (pulse energy) and scanning speed. During our experimental study, we have actually conducted DoE analyses and particularly selected laser power as 50%, 55%, 60%, 65%, 70%, 75%, 80%, 85%; scanning speed as 1500 mm/s, 2000 mm/s, 2500 mm/s, 3000 mm/s, 3500 mm/s, 4000 mm/s, 4500 mm/s, 5000 mm/s; scanning times as 1, 2, 3, 4. According to L_64_(8^9^) the orthogonal table, we have totally carried out 64 process parameters. The optimum process condition for surface quality and properties according to the orthogonal experiment is reported. When laser power is set to 65%, scanning speed of the laser beam is 3000 mm/s, laser frequency is 100 kHz, scanning interval is 20 μm. The laser intensity was 6.75 × 10^6^ W/cm^2^. The grinded process experiments are performed in an Ar shields environment.

To examine the surface topography, the sample surface was observed using a 3D laser scanning confocal microscope (LSCM, VK-X100, KEYENCE, Osaka, Japan) and scanning electron microscope (SEM, GEMINISEM 500, ZEISS, Oberkochen, Germany) equipped with an energy dispersive spectrometer (EDS). The surface Ra are evaluated by surface profilometers (MahrSurf M 300 C, Mahr, Göttingen, Germany). Phase composition of silicon wafer was determined by X-ray diffractometer (XRD, D/max2200PC, Rigaku, Tokyo, Japan). A laser micro-Raman spectrometer (inVia Reflex, Renishaw, Gloucestershire, UK) was used to examine the surface crystal structure. The laser wavelength of the spectrometer was 532 nm. The room-temperature *I-V* characteristic curves and resistivity of single-crystal silicon wafers were tested by four-probe method using an ZEM-3 system (Ulvac RIko, Chigasaki, Kanagawa, Japan) instrument to obtain the influence of laser irradiation on electrical properties of silicon wafer without the effect of probe position on the result.

## 3. Result and Discussion

### 3.1. Effect of Laser Grinding on Surface Morphology

The surfaces morphology of the as-revised and laser-grinded single-crystal silicon wafer samples are shown in Figure 2a. Ra is the arithmetic mean of the surface roughness profile; Rz is the height of the irregularities of the surface roughness profile at 10 points. The left half of Figure 2a shows the surface morphology of the as-received surface and surface roughness values Ra and Rz were 0.4 μm and 2.47 μm, respectively. The original surface exhibits a rough appearance. Numerous small embossments and some grooves can be seen on the surface. This indicated that the smooth surface could not be obtained after mechanical sawing. The right half of Figure 2a shows the surface morphology of the laser-grinded surface. After laser grinding, surface roughness Ra and Rz value were reduced to 0.075 μm and 0.34 μm, respectively. It was immediately apparent that following laser grinding there was a large improvement in surface quality, with the grinded regions being significantly smoother compared to the as-received surface. This phenomenon suggests that the embossment and groove are melted and incorporated during laser grinding.

The top images of Figure 2b-1 and Figure 2b-2 illustrate the three-dimensional (3D) surface topographies of the as-received surface and laser-grinded surface, respectively. The 3D profile of the samples within a measurement region of 200 μm × 290.3 μm of the top surface. The red arrow in Figure 2b-2 represents the laser grinding direction. Abundant peaks are observed on the as-received surface, and, after laser grinding, the number peaks are reduced and the surface is smoothed. The maximum peak height is reduced from 14 μm to 5 μm. However, ripples on the samples used in this work can be observed along the direction of the scan track of the laser. The bottom images of Figure 2b-1 and Figure 2b-2 are the two-dimensional (2D) surface profiles of the as-received surface and laser-grinded surface, respectively. The linear surface profiles of the samples, which have a length of 200 μm, are taken from lines perpendicular to the laser-grinding direction. The surface profiles of the as-received surface fluctuates between 9.21 μm and 14.8 μm, whereas that of laser grinding fluctuates between 2.66 μm and 4.8 μm.

The mechanism of laser grinding includes two steps, which are shown in Figure 3a,b. The first step is laser directly removal of scratches and oxides caused by sawing process. The laser beam with high energy density is absorbed by the surface oxide layer, which forms a rapidly expanding plasma and produces shock wave. The shock wave causes the pollutants to fragment and be removed [19]. While the next step is laser melting due to thermal accumulation. Figure 3c shows that the surface of silicon wafer melts during laser grinding. Molten silicon is redistributed in flow due to the simultaneous action of surface tension and Maragani effect [20].

### 3.2. Effect of Laser Grinding on Microstructure Evolution

To investigate the microstructure change of single-crystal silicon wafer after laser grinding, the as-received surface and laser-grinded surface were exposed by SEM and EDS. As shown in Figure 4, the EDS face scanning analyses clearly revealed Si, C, and O are found in two surfaces. Si element are enriched and remain mostly homogeneous across the as-received surface and laser-grinded surface. However, the content of C and O in laser grinded surface reduced compared to the as-received surface. The decrease of C and O in laser-grinded surface results from oxide layer and pollutant disappearing by laser ablation.

XPS analysis was used to identify the chemical nature of the single-crystal silicon wafer surface before and after surface modification. XPS spectra obtained for as-received and laser-grinded surface are provided in Figure 5. XPS spectrum of both surfaces showed the presence of peaks corresponding to C1s, O1s, and Si2p. Figure 5a show the C1s spectra of the as-received and laser-grinded surface. A strong peak corresponding to the C-C can be observed in C1s spectra of the as-received surface. However, the intensity of the C-C peak obviously decreased in C1s spectra of the laser-grinded surface. Other weaker peak C=O was also found in the C1s spectra. The intensity of C=O peak was stronger in C1s spectra of as-received surface than that in C1s spectra of laser-grinded surface. These results indicate that the oxide layer was sufficiently removed after laser grinding [21]. The conclusion can also be proved by Si-O and C=O peaks variation in O1s spectrum: the stronger Si-O peak in single-crystal silicon original surface decreased after laser grinding (Figure 5b). Meanwhile, others peaks disappeared. It indicates that fewer or even no C=O and Si-O bonds were present on the laser-grinded surface and the disappeared oxide layer may be SiO_2_ [22]. The same conclusion can be concluded in Figure 5c. As shown in Figure 5c, peaks corresponding to Si-O and Si were relatively strong on as-received surface compared to those of the laser-grinded surface in Si2p spectrum. The peak at 100 eV is weaker after laser grinding, which indicates that the content of Si-C decreases after grinding. This is in good agreement with EDS results due to the reduction of C content. In addition, SiO_2_ peak disappeared from the laser-grinded surface. The result proved that the oxide layer is SiO_2_ [23,24].

The XRD patterns of as-received surface and laser-grinded surface of single-crystal silicon wafer samples are depicted in Figure 6a. As shown in Figure 6a, as-received surface and laser-grinded surface exhibit a sharp diffraction peak Si (400) at 2θ ≈ 69.13° indicating that Si is the matrix phase. Apart from the strong diffraction peaks corresponding to the Si matrix, two minor peaks appear at 2θ ≈ 32.985° and 61.75° corresponding to the SiO_2_ (440) and SiO_2_ (−181) in the as-received surface, respectively. Obvious peak change can be observed after laser grinding. Only a minor peak at 2θ ≈ 61.75° existed in the laser-grinded surface and the peak intensity appreciably declined. It is reasonable to conclude that such a secondary phase still exists in the laser-grinded surface, although present in a very small volume fraction. Furthermore, no significant deviation in the peak position is observed in Figure 6a. As mentioned above, conclusions can be reached that the volume fraction of the SiO_2_ phase significantly decreased during the laser grinding process. This phenomenon is mainly attributed to the decrease in the oxygen content and no more oxygen can be injected into the silicon wafer during the laser-grinded process in Ar environment. This can be supported by the conclusion of EDS analysis.

The crystallinity of the as-received surface and laser-grinded surface in single-crystal silicon were also investigated using Raman spectroscopy, as shown in Figure 6b. The as-received surface and laser-grinded surface respectively showed a sharp peak at 520.29 cm^−1^ and 520.016 cm^−1^, which both are close to the bare single-crystal silicon (100) peak at 520 cm^−1^. Moreover, a minor peak was presented in the Raman spectroscopy of as-received surface. It is known that Raman spectra of the amorphous silicon, polycrystalline silicon, and single-crystal silicon are located around 480 cm^−1^, 500–515 cm^−1^, and 520 cm^−1^, respectively [25]. Therefore, the minor peak represented polycrystalline silicon. The conclusion indicates that the polycrystalline silicon was completely transformed into single-crystal silicon during the laser-grinding process. Generally, conventional grinding will cause polycrystalline silicon layers [4,11,13]. After a laser pulse is irradiated on the surface, the polycrystalline silicon layers will be melted and becomes thicker and thicker when laser irradiation continues, reaching the whole polycrystalline silicon layers. Then after the laser pulse, cooling will result in bottom-up epitaxial regrowth from the single-crystal silicon substrate. In this way, a perfect single-crystal structure can be obtained in the laser-irradiated surface [12].

### 3.3. Effect of Laser Grinding on Resistivity

Figure 7 shown the *I-V* characteristic curve of as-receive and laser-grinded of single-crystal silicon wafer. The current linearly increases with voltage increasing. The *I-V* slope of laser-grinded is slightly smaller than that of as-received.

The electrical resistivity of single-crystal silicon wafer was measured at room temperature by using four-probe test, and the resistivity was calculated via the following equation:(1)$$\rho =\frac{R\times L}{S}$$
where *ρ* is the resistivity, *R* is the resistance of silicon wafer sample, *S* is the cross-sectional area of testing samples, and *L* is the distance between the two electrodes.

The regression analysis of *I-V* characteristic curve is used to estimate *R* and *ρ*. The *R* before and after laser grinding are estimated as 0.281 Ω and 0.266 Ω, respectively, while *ρ* is 0.572 Ω·cm and 0.417 Ω·cm, respectively. The average value of the standardized residual error of these two regression models is 0, and the deviation is 0.967, which is close to 1. It is proved that the *R* and *ρ* values obtained by using the *I*-*V* characteristic curve are unbiased.

Table 1 lists the resistivity of silicon wafer with as-received and laser-grinded. Compared with the resistivity of as-received silicon wafer, the resistivity of laser-grinded silicon wafer was smaller. The decrease of the resistivity is attributed to the disappeared polycrystalline silicon, as shown in Figure 4b. Therefore, the laser grinding process can reduce the resistivity of silicon wafer.

## 4. Conclusions

In this study, surface quality, microstructure, and resistivity of single-crystal silicon wafer before and after laser grinding were analyzed. The following conclusions can be drawn:
Laser grinding reduced the surface roughness of single-crystal silicon wafer by over 81%, when measured at the mm scale, but achieved a roughness level of only Ra = 75 nm when measured at the micro-scale. Meanwhile, the laser-grinded surface changes were smother because laser grinding removes abundant peaks.Laser grinding could decline oxygen content of the single-crystal silicon wafer. EDS and XPS analysis results proved this conclusion. XRD analysis showed that the SiO_2_ phase disappeared after laser grinding. It indicated that the oxide layer was completely removed during the laser-grinding processing.Raman spectra of as-received surface and laser-grinded surface analysis showed that crystallinity in the grinded region were changed after laser grinding. Polycrystalline Si was transformed into the single- crystal structure during laser grinding.Laser grinding could reduce resistivity of single-crystal silicon wafer due to disappeared polycrystalline silicon, but the effect is not obvious.

## Figures and Tables

**Figure 1 micromachines-12-00262-f001:**
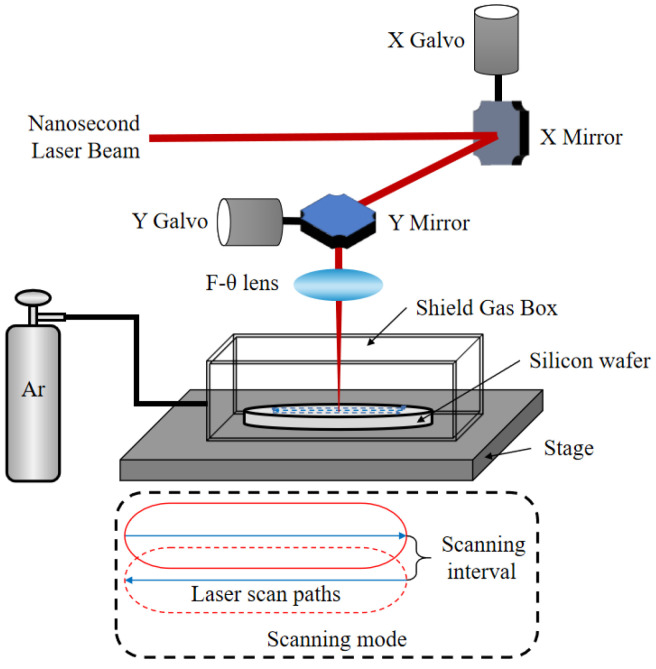
Schematic for experimental setup of laser grinding single-crystal silicon wafer.

**Figure 2 micromachines-12-00262-f002:**
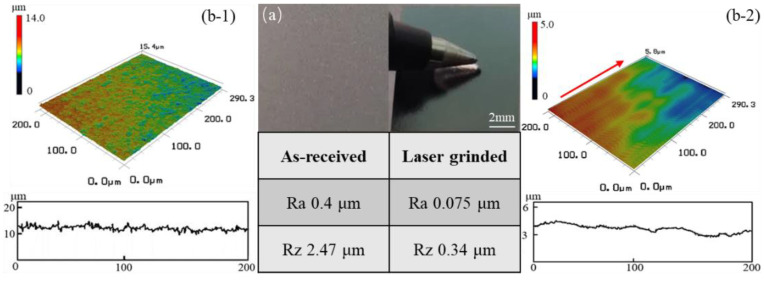
Surface topography: (**a**) as-received and laser-grinded surfaces at macro-scale; (**b-1**) 3D topographic image of as-received surface, (**b-2**) 3D topographic image of the laser-grinded surface.

**Figure 3 micromachines-12-00262-f003:**
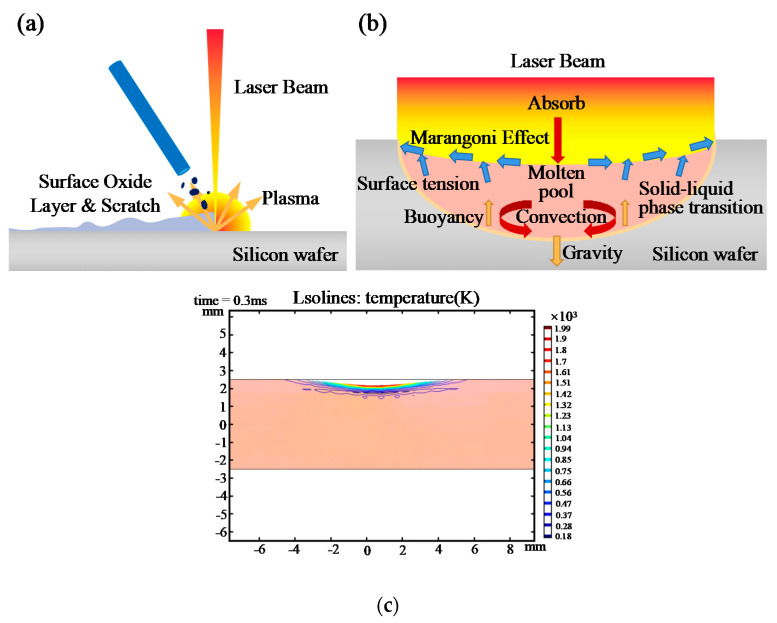
Schematic diagram of laser grinding mechanism: (**a**) laser directly removal of scratches and oxides, (**b**) laser melting, (**c**) temperature field simulation.

**Figure 4 micromachines-12-00262-f004:**
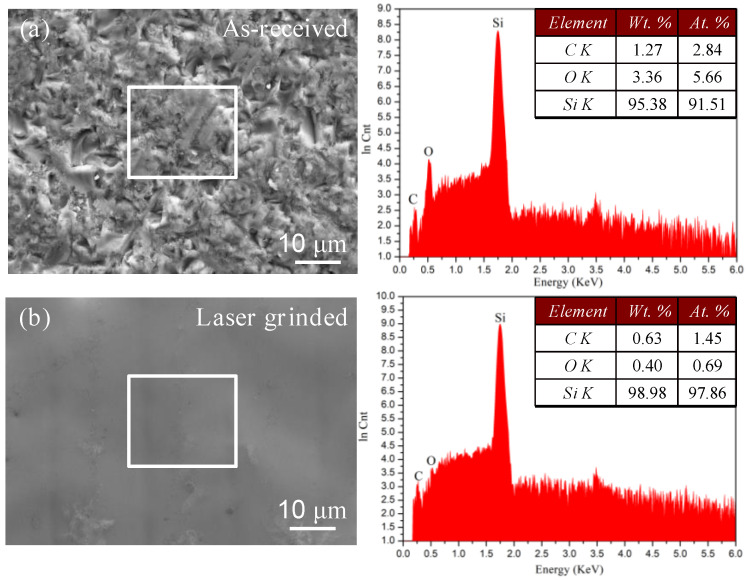
Elemental distribution and chemical composition (*Wt*. %/*At*. %) correspond to the white box in SEM from EDS spectrum analysis.

**Figure 5 micromachines-12-00262-f005:**
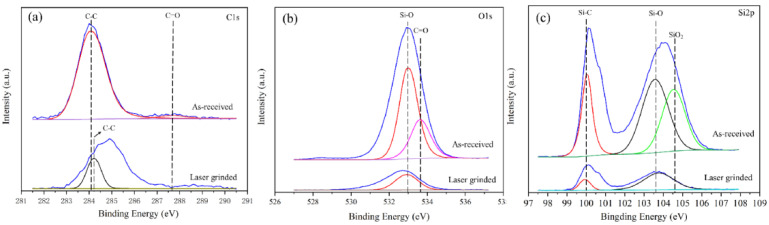
XPS spectra of the as-received and laser polished surface, which correspond to (**a**–**c**) result of curve fitting of C1s, O1s, and Si2p with bulk single crystal silicon surfaces (The XPS curves have been extracted from raw data after measurement and proceeded by standard analyses via Avantage software 5.967 (Thermo Fisher Scientific, Waltham, MA, USA)).

**Figure 6 micromachines-12-00262-f006:**
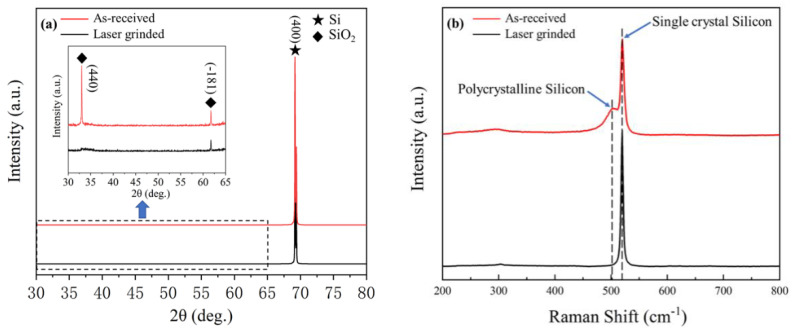
(**a**) XRD and (**b**) Raman spectrums from the laser-grinded surface and the as-received surface.

**Figure 7 micromachines-12-00262-f007:**
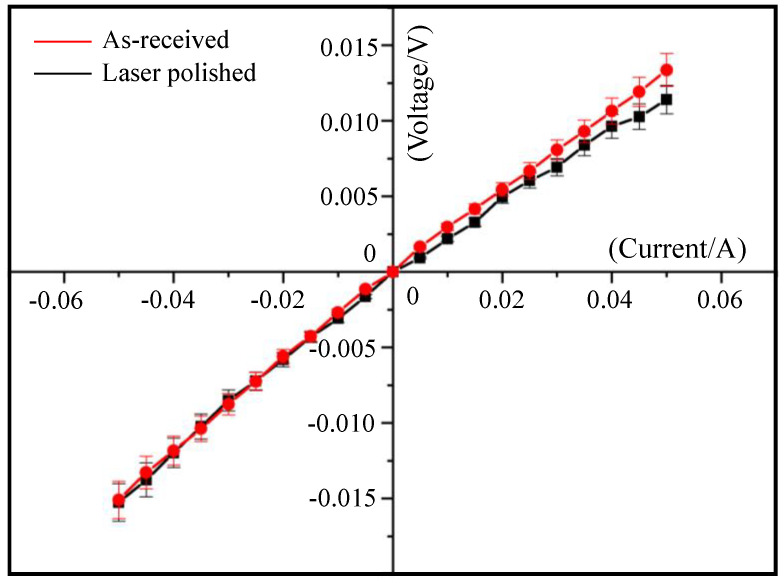
Comparison of *I-V* characteristic curves of as-received and laser-grinded surfaces at the single-crystal silicon wafer.

**Table 1 micromachines-12-00262-t001:** The resistivity of single-crystal silicon wafer samples with as-received and laser-grinded.

Specimen	As-Received	Laser-Grinded
Resistivity (Ω·cm)	0.572	0.417

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
