# Peer review of "Laser Grinding of Single-Crystal Silicon Wafer for Surface Finishing and Electrical Properties"

_micromachines, 2021, doi:10.3390/mi12030262_

Round 1

Reviewer 1 Report

This paper reports laser grinding process to improve the surface quality of the Si wafer after diamond sawing. The procedure from experiment to analysis are reasonable and logical, and the results of this study are also very useful in that the laser grinding process can effectively enhance the quality of the mechanically machined Si surface. Therefore, I recommend this manuscript to be published in micromachines, but before the publication, following items should be properly revised and answered.

  1. Please insert the schematics for the experimental setup to enhance the understanding for the laser grinding process
  2. Need to explain briefly the theory of the laser grinding process in the text. How does the laser irradiation improve the surface quality ? (better to use the figure to explain the mechanism)
  3. Please describe or explain the difference between Ra and Rz (or add definition of them in the text)
  4. Any reason to use just one process condition ? (is this optimum process condition ? ) Depending on the process parameters, the surface quality will be different. I wonder if the result shown in the paper is only for the test by one process condition.
  5. Beam intensity information is also very important in laser material processing. Please describe beam intensity condition used for the process

Author Response

Reply to Reviewer’s Comments

Micromachines

Manuscript ID: micromachines-1110046

Title: Laser Grinding of Single-crystal Silicon Wafer

Authors: Yimeng Wang, Xinxin Li, Yingchun Guan

We would like to express our sincere gratitude to both the Editor’s timely handling our manuscript and the Reviewers’ invaluable suggestions which significantly improve the quality of the manuscript. We have thoroughly modified the manuscript according to the reviewer’s comments as listed below.

Notes: All the modifications are labelled in BLUE in the revised manuscript. Also in the authors’ reply to the reviewers, the paper numbers, figure numbers, and references are related to the revised manuscript.

Reviewers' comments:

Reviewer 1: This paper reports laser grinding process to improve the surface quality of the Si wafer after diamond sawing. The procedure from experiment to analysis are reasonable and logical, and the results of this study are also very useful in that the laser grinding process can effectively enhance the quality of the mechanically machined Si surface. Therefore, I recommend this manuscript to be published in micromachines, but before the publication, following items should be properly revised and answered.

  1. Please insert the schematics for the experimental setup to enhance the understanding for the laser grinding process.

Reply: Thank you for your suggestions. The schematic for the experimental setup of laser grinding silicon wafer as Figure 1 has been added on Page 3 as follows.

Figure 1. Schematic for experimental setup of laser grinding single-crystal silicon wafer.

  1. Need to explain briefly the theory of the laser grinding process in the text. How does the laser irradiation improve the surface quality? (better to use the figure to explain the mechanism)

Reply: The theory of the laser grinding process is important to justify, and it has been provided in Figure 3 on Page 4 as followings.

“The mechanism of laser grinding includes two steps. The first step is laser directly removal of scratches and oxides caused by sawing process. The laser beam with high energy density has been absorbed by the surface oxide layer, which forms a rapidly expanding plasma and produces shock wave [19]. The shock wave causes the pollutants to fragment and be removed. While the next step is laser melting due to thermal accumulation. Molten silicon is redistributed in flow due to the simultaneous action of surface tension and Maragani effect [20].

Figure 3. Schematic diagram of laser grinding mechanism: (a) laser directly removal of scratches and oxides, (b) laser melting, (c) temperature field simulation.

  1. Please describe or explain the difference between Ra and Rz (or add definition of them in the text).

Reply: The definition of Ra and Rz has been added in the Paragraph 2 on Page 3 as follows.

“Ra is the arithmetic mean of the surface roughness profile, while Rz is the height of the irregularities of the surface roughness profile at 10 points.”

  1. Any reason to use just one process condition? (is this optimum process condition?) Depending on the process parameters, the surface quality will be different. I wonder if the result shown in the paper is only for the test by one process condition.

Reply: During our experimental study, we have actually conducted DoE analyses[R1] and particularly selected laser power as 50%, 55%, 60%, 65%, 70%, 75%, 80%, 85%; scanning speed as 1500 mm/s, 2000 mm/s, 2500 mm/s, 3000 mm/s, 3500 mm/s, 4000 mm/s, 4500 mm/s, 5000 mm/s; scanning times as 1, 2, 3, 4, as shown as L64(89) orthogonal table. According to the orthogonal table, we have totally carried out 64 process parameters. What is reported in this manuscript is the optimum process condition for surface quality and properties according to the orthogonal experiment.

  1. Beam intensity information is also very important in laser material processing. Please describe beam intensity condition used for the process.

Reply: The beam intensity condition used for the process has been added in the Paragraph 4 on Page 2 as follows. “The laser intensity was 6.75×106w/cm2.”

Reference

  • Jiaru Zhang, Kai Guan, Zhen Zhang, Yingchun Guan. In Vitro Evaluation of Ultrafast Laser Drilling Large-size Hole on Sheepshank Bone. OPTICS EXPRESS. 28(17), 2020, 25528.

Reviewer 2 Report

comments reported in the pdf file

Author Response

Reply to Reviewer’s Comments

Micromachines

Manuscript ID: micromachines-1110046

Title: Laser Grinding of Single-crystal Silicon Wafer

Authors: Yimeng Wang, Xinxin Li, Yingchun Guan

We would like to express our sincere gratitude to both the Editor’s timely handling our manuscript and the Reviewers’ invaluable suggestions which significantly improve the quality of the manuscript. We have thoroughly modified the manuscript according to the reviewer’s comments as listed below.

Notes: All the modifications are labelled in BLUE in the revised manuscript. Also in the authors’ reply to the reviewers, the paper numbers, figure numbers, and references are related to the revised manuscript.

Reviewers' comments:

Reviewer 2: The paper “Laser Grinding of Single-crystal Silicon Wafer” by Xinxin Li and Yingchun Guan shows the advantages of laser grinding for the treatment of boron-doped single-crystal silicon wafer machined by diamond sawing. Specifically, monitoring the roughness, the carbon atoms concentrations, the oxide layer and the resistivity, they show an improvement in the quality of the silicon surface. The relevance of surface quality of silicon wafer for electronic application can motivate the publication of the manuscript in Micromachines. However, before the publication the authors should solve some technical issues reported below:

  1. In the first line of the abstract the authors mention that the work is performed in boron doped

single crystal silicon wafer. However, in the manuscript I cannot find any reference to doping in the manuscript. The authors should quantify the doping. Could they find some experimental evidence of the doping? Is it modified by laser grinding?

Reply: The present doping boron single-crystal silicon has been produced by Suzhou Yancai Nanomicro technology Co., Ltd, where the doping boron is used to obtain p-type wafer and the doping concentration of boron atom is 1016/cm3. Since the density of single-crystal silicon is about 2.32 g/cm3, the mass fraction of boron per unit volume is nearly 4×10-7, which is extremely small. Therefore, this study mainly has been focused on the improvement of surface quality after laser grinding including the main elements as Si, O, and C contents. The change of boron element will be studied in our further experiment.

  1. Pag.2 line 74: The authors should explain what they mean with scanning interval.

Reply: Please refer to Figure 1 on Page 3 as follows.

Figure 1. Schematic for experimental setup of laser grinding single-crystal silicon wafer.

  1. The authors should report the time of the laser grinding process for cm2. Considering the scanning speed and a step between lines of 30 μm, I calculate that you need ~15 minutes for 1 cm2. Is it right?

Reply: We actually have conducted 2-3 seconds for 1 cm2 of laser grinding process, which is mainly determined by scanning speed, scanning interval and focal beam diameter.

  1. Figure 2: the x-axes of EDS spectra have Chinese characters, and the labels of y axis are missing.

Reply: Sorry for the typo. We have been modified the labels of y axis in the Line 154 on Page 5 as follows.

Figure 4. Elemental distribution and chemical composition (Wt. %/At. %) correspond to the white box in SEM from EDS spectrum analysis.

  1. Figure 2: The carbon and oxygen peaks are barely visible. The authors should enhance the visibility of peak. Maybe a log scale in y-axis can help.

Reply: Thank you for your suggestions. We are not able to perform the log scale in y-axis since our EDS data is not able to be edited. In order to present the detailed content of carbon and oxygen element, we have provided the quantified data in Figure 4 based on the EDS spectra.

  1. Figure 3: There are smooth lines above the experimental data. I suppose they represent the fitted peak contributions used to reproduce the experimental data. However, in Figure3b the peak related to C=O is not shown. The authors should comment these lines in the caption. Moreover, the “(a)” label is missing in the caption.

Reply: Sorry for the typo. The peak related to C=O is not shown at binding energy (532.4eV) according to NIST Standard Reference Database 20, Version 4.1[R2]. Moreover, we have modified the Figure3b and added the “(a)” label and comments of these lines in the caption of Figure3 in revised manuscript in the Line 178 on Page 6 as follows.

Figure 5. XPS spectra of the as-received and laser polished surface, which correspond to (a-c) result of curve fitting of C1s, O1s and Si2p with bulk single crystal silicon surface.

  1. Figure 3c: The authors should explain, why the peak at 100 eV assigned to Silicon is weaker after laser grinding.

Reply: Sorry for the typo. The peak at 100 eV refers to Si-C according to NIST Standard Reference Database 20, Version 4.1, which has been formed due to the contamination of the ambient[R2]. The peak at 100 eV is weaker after laser grinding, which indicates that the content of Si-C decreases after grinding. This is in good agreement with EDS results due to the reduction of C content.

  1. Pag.5 lines 157-158: “It is reasonable to conclude that such a secondary phase, although present in a very small volume fraction.” A verb is missing.

Reply: Sorry for the typo. We have been modified this sentence in the Paragraph 1 on Page 6 as follows. “It is reasonable to conclude that such a secondary phase still exists in the laser grinded surface, although present in a very small volume fraction.”

  1. Pag 6 lines 197-198: “Compared with the resistivity of as-received silicon wafer, the resistivity of laser grinded silicon wafer smaller”. Figure 5 show similar profile for the two sample, then, a detailed error estimation of R and ρ should be performed to extract this information.

Reply: The regression analysis of I-V characteristic curve is used to estimate R and ρ. The R before and after laser grinding are estimated as 0.281 Ω and 0.266 Ω, respectively, while ρ is 0.572 Ω·cm and 0.417 Ω·cm, respectively. The average value of the standardized residual error of these two regression models is 0, and the deviation is 0.967, which is close to 1. It is proved that the R and ρ values obtained by using the I-V characteristic curve are unbiased.

  1. The EDS results reported in the tables of Figure1 shows a decreasing in the oxygen concentration of >97%. On the contrary, the XPS intensities of SiO and SiO2 peaks decrease of a factor ≲ 85%. The authors should comment this mismatch.

Reply: Thank you for your comments. It is noted that both EDS and XPS are semi quantitative methods for chemical analysis. On one hand, the peak intensity does not have a strict corresponding relationship with the actual content. On the other hand, the change of peak intensity can not strictly reflect the change of actual content. Moreover, EDS can detect the area several microns below the surface of the sample, while XPS mainly detects the area 3-5 nm below the surface of the sample; Finally, XPS is more sensitive to light weight elements than that of EDS. Correspondingly, these two results are shown as the mismatch.

Reference

  • Jiaru Zhang, Kai Guan, Zhen Zhang, Yingchun Guan. In Vitro Evaluation of Ultrafast Laser Drilling Large-size Hole on Sheepshank Bone. OPTICS EXPRESS. 28(17), 2020, 25528.
  • NIST Chemistry Web Book, NIST Standard Reference Database 20.

Reviewer 3 Report

The reviewer comments of the paper «Laser Grinding of Single-crystal Silicon Wafer»- Reviewer

The authors presented an article «Laser Grinding of Single-crystal Silicon Wafer». However, there are several points in the article that require further explanation.

Comment 1:

  1. Materials and Methods

Specify what roughness parameters were measured: Ra is the arithmetic mean of the surface roughness profile; Rz is the height of the irregularities of the surface roughness profile at 10 points.

Give a photo of the laser grinding machine where the processing diagram is visible. Show the main elements.

Explain why you have chosen the modes of laser grinding.

How many repetitions of the experiment were used in the studies?

Comment 2:

  1. Result and discussion

The quality of the all figures and their resolution should be greatly improved. Are all figures original? If not needed appropriate citations and permissions.

What is the hardness of the workpiece and how was it measured?

Comment 3:

Conclusions It is important to indicate for which modes of laser grinding and for which material these conclusions were obtained.

In addition, it is necessary to more clearly show the novelty of the article and the advantages of the proposed method. What is the difference from previous work in this area? Show practical relevance. Conclusions should reflect the purpose of the article.

Comment 4:

In the title of the article, it is preferable to briefly designate the parameters under study: surface integrity.

Comment 5:

Indicate in the abstract the modes of laser grinding for which the research results were obtained.

The article is interesting and helpful. However, the authors did only one experiment which makes the content somewhat "poor". Although, in general, the study is comprehensively done. Authors should carefully study the comments and make improvements to the article step by step. Mark all changes in color. After major changes can an article be considered for publication in the "Micromachines".

Author Response

Reply to Reviewer’s Comments

Micromachines

Manuscript ID: micromachines-1110046

Title: Laser Grinding of Single-crystal Silicon Wafer

Authors: Yimeng Wang, Xinxin Li, Yingchun Guan

We would like to express our sincere gratitude to both the Editor’s timely handling our manuscript and the Reviewers’ invaluable suggestions which significantly improve the quality of the manuscript. We have thoroughly modified the manuscript according to the reviewer’s comments as listed below.

Notes: All the modifications are labelled in BLUE in the revised manuscript. Also in the authors’ reply to the reviewers, the paper numbers, figure numbers, and references are related to the revised manuscript.

Reviewers' comments:

Reviewer 3: The authors presented an article «Laser Grinding of Single-crystal Silicon Wafer». However, there are several points in the article that require further explanation.

Comment 1:

Materials and Methods

  1. Specify what roughness parameters were measured: Ra is the arithmetic mean of the surface roughness profile; Rz is the height of the irregularities of the surface roughness profile at 10 points.

Reply: Both Ra and Rz have been measured by the surface profilometer in this manuscript.

  1. Give a photo of the laser grinding machine where the processing diagram is visible. Show the main elements.

Reply: The schematic for the experimental setup has been added on Page 3. Please refer Reply 1 to Reviewer 1.

  1. Explain why you have chosen the modes of laser grinding.

Reply: In order to meet the demanding requirements of IC industry for semiconductor materials with high precision and ultra smooth surface, a high-precision, efficient, good repeatability and reliable wafer grinding method causes much attentions recently. Laser melting method has been useful for surface smoothing of metallic metals in our group, and we would like to further explore it for Silicon wafer application for curiosity at the first place. After several trial and errors, we have successfully conducted laser smoothing for mechanical-machined wafer, and named it as laser grinding.

  1. How many repetitions of the experiment were used in the studies?

Reply: We have totally carried out 64 repetitions of the experiment were used in the studies. Please refer Reply 4 to Reviewer 1.

Comment 2:

Result and discussion

  1. The quality of the all figures and their resolution should be greatly improved. Are all figures original? If not needed appropriate citations and permissions.

Reply: All figures are original, and we have provided high quality of the all figures in the revised manuscript.

  1. What is the hardness of the workpiece and how was it measured?

Reply: Single crystal Si can be indented on the (001) face using an instrumented Berkovich indenter (Nanotest 600, Micromaterials, UK) with loads ranging between 20 mN and 100 mN, and the measured hardness at room temperature is 13 ± 1 GPa [R3].

Comment 3:

  1. Conclusions It is important to indicate for which modes of laser grinding and for which material these conclusions were obtained. In addition, it is necessary to more clearly show the novelty of the article and the advantages of the proposed method. What is the difference from previous work in this area? Show practical relevance. Conclusions should reflect the purpose of the article.

Reply: Please refer to our previous Reply, and these conclusions can be further applied to metals, ceramics, glass, etc[R4-R6]. The novelty of this paper is that we have firstly reported laser grinding method for good surface quality of silicon wafer. The proposed process is different from conventional methods, because it can completely remove oxide layer and avoid damage [R4-R6]. In this study, laser grinding has reduced surface roughness of single-crystal silicon wafer by over 81% with the grinding speed is 2.2 s/cm2, which is much more efficient than traditional grinding process[R7].

Comment 4:

  1. In the title of the article, it is preferable to briefly designate the parameters under study: surface integrity.

Reply: Thank you for constructive comments. We have modified the revised title as “Laser Grinding of Single-crystal Silicon Wafer for Surface Finishing and Electrical Properties”.

Comment 5:

  1. Indicate in the abstract the modes of laser grinding for which the research results were obtained.

The article is interesting and helpful. However, the authors did only one experiment which makes the content somewhat "poor". Although, in general, the study is comprehensively done. Authors should carefully study the comments and make improvements to the article step by step. Mark all changes in color. After major changes can an article be considered for publication in the "Micromachines".

Reply: We are very grateful to the reviewers for the valuable comments. We have carefully considered the comments and made modifications in the manuscript accordingly as listed below. All the modifications are shown in BLUE in the revised manuscript.

Reference

  • Jiaru Zhang, Kai Guan, Zhen Zhang, Yingchun Guan. In Vitro Evaluation of Ultrafast Laser Drilling Large-size Hole on Sheepshank Bone. OPTICS EXPRESS. 28(17), 2020, 25528.
  • NIST Chemistry Web Book, NIST Standard Reference Database 20.
  • J. Vandeperre, F. Giuliani, S.J. Lloyd, W.J. Clegg, The hardness of silicon and germanium. Acta Materialia. 55(18), 2007, 6307-6315.
  • Seungjong Lee, Zabihollah Ahmadi, Jonathan W. Pegues, Masoud Mahjouri-Samani, Nima Shamsaei. Laser polishing for improving fatigue performance of additive manufactured Ti-6Al-4V parts. Optics & Laser Technology. 134, 2021, 106639.
  • Ximin Zhang, Lingfei Ji, Litian Zhang, Wenhao Wang, Tianyang Yan. Polishing of alumina ceramic to submicrometer surface roughness by picosecond laser. Surface and Coatings Technology. 397, 2020, 125962.
  • Jörg Hildebrand, Kerstin Hecht, Jens Bliedtner, Hartmut Mü Laser Beam Polishing of Quartz Glass Surfaces. Physics Procedia. 12, Part A, 2011, 452-461.
  • Pei Z, Fisher GR, Liu J. Grinding of silicon wafers: A review from historical perspectives. International Journal of Machine Tools and Manufacture. 2008, 48, 1297-1307.

Round 2

Reviewer 2 Report

see pdf file

Author Response

Reply to Reviewer’s Comments

Micromachines

Manuscript ID: micromachines-1110046

Title: Laser Grinding of Single-crystal Silicon Wafer

Authors: Yimeng Wang, Xinxin Li, Yingchun Guan

We would like to express our sincere gratitude to both the Editor’s timely handling our manuscript and the Reviewers’ invaluable suggestions which significantly improve the quality of the manuscript. We have thoroughly modified the manuscript according to the reviewer’s comments as listed below.

Notes: All the modifications are labelled in BLUE in the revised manuscript. Also in the authors’ reply to the reviewers, the paper numbers, figure numbers, and references are related to the revised manuscript.

Reviewers' comments:

Reviewer 2:

The paper “Laser Grinding of Single-crystal Silicon Wafer” by Xinxin Li and Yingchun Guan has been largely improved by the authors. However, few issues are not addressed in an appropriate way.

The authors should consider these comments before publication. The remaining issues are reported below:

  1. In the first line of the abstract the authors mention that the work is performed in boron doped single crystal silicon wafer. However, in the manuscript I cannot find any reference to doping in the manuscript. The authors should quantify the doping. Could they find some experimental evidence of the doping? Is it modified by laser grinding?

Reply: The present doping boron single-crystal silicon has been produced by Suzhou Yancai Nanomicro technology Co., Ltd, where the doping boron is used to obtain p-type wafer and the doping concentration of boron atom is 1016/cm3. Since the density of single-crystal silicon is about

2.32 g/cm3, the mass fraction of boron per unit volume is nearly 4×10-7, which is extremely small. Therefore, this study mainly has been focused on the improvement of surface quality after laser grinding including the main elements as Si, O, and C contents. The change of boron element will be studied in our further experiment.

Reviewer reply: If the authors does not investigate the doping, they should avoid mentioning the doping in the abstract. The doping and the relative concentration can be described in detail in the section “Materials and Methods”.

Reply: Thanks for constructive suggestion. We have deleted the “ boron-doped” in the first sentence of the abstract. Moreover, the doping and the relative concentration has been added in the Paragraph 3 on Page 2 as follows: “with the boron doping concentration of 1016/cm3”.

  1. Figure 2: The carbon and oxygen peaks are barely visible. The authors should enhance the visibility of peak. Maybe a log scale in y-axis can help.

Reply: Thank you for your suggestions. We are not able to perform the log scale in y-axis since our EDS data is not able to be edited. In order to present the detailed content of carbon and oxygen element, we have provided the quantified data in Figure 4 based on the EDS spectra.

Reviewer reply: The authors quantified data, but they do not have access to data to perform editing. Maybe I did not understand the difficulty, but this answer seems very strange to me. This is a minor point, nevertheless, I expect an effort from the authors to improve the readability of the figure.

Reply: We have carried out another EDS measurement recently, and performed the logarithm of y-axis according to your previous suggestion on Page 5 as follows.

Figure 4. Elemental distribution and chemical composition (Wt. %/At. %) correspond to the white box in SEM from EDS spectrum analysis.

  1. Figure 3: There are smooth lines above the experimental data. I suppose they represent the fitted peak contributions used to reproduce the experimental data. However, in Figure3b the peak related to C=O is not shown. The authors should comment these lines in the caption. Moreover, the “(a)” label is missing in the caption.

Reply: Sorry for the typo. The peak related to C=O is not shown at binding energy (532.4eV) according to NIST Standard Reference Database 20, Version 4.1[R2]. Moreover, we have modified the Figure3b and added the “(a)” label and comments of these lines in the caption of Figure3 in revised manuscript in the Line 178 on Page 6 as follows.

Figure 5. XPS spectra of the as-received and laser polished surface, which correspond to (a-c) result of curve fitting of C1s, O1s and Si2p with bulk single crystal silicon surface.

Reviewer reply: The description of the curves in the caption is missing. The authors should add it, explaining the method to extract these curves for a not expert reader.

Reply: Thank you for your comments. We have added the description of the curves in the caption of Figure 5 as followings: “Figure 5. XPS spectra of both as-received and laser-polished surface, which correspond to (a-c) result of curve fitting of C1s, O1s and Si2p with bulk single crystal silicon surface (The XPS curves have been extracted from raw data after measurement and proceeded by standard analyses via Avantage software).”

  1. Pag 6 lines 197-198: “Compared with the resistivity of as-received silicon wafer, the resistivity of laser grinded silicon wafer smaller”. Figure 5 show similar profile for the two sample, then, a detailed error estimation of R and ρ should be performed to extract this information.

Reply: The regression analysis of I-V characteristic curve is used to estimate R and ρ. The R before and after laser grinding are estimated as 0.281 Ω and 0.266 Ω, respectively, while ρ is 0.572 Ω·cm and 0.417 Ω·cm, respectively. The average value of the standardized residual error of these two regression models is 0, and the deviation is 0.967, which is close to 1. It is proved that the R and ρ values obtained by using the I-V characteristic curve are unbiased.

Reviewer reply: The authors report the error related to regression analysis, but they do not report the experimental errors due to the uncertainty in the S, L, and ρ values. They should support the results with an experimental error estimation. Are the experimental error bars considered in the regression analysis?

Reply: We have added the experimental error bar to Figure 7 accordingly on Page 7 as follows.

Figure 7. Comparison of I-V characteristic curves of as-received and laser grinded surfaces at the single-crystal silicon wafer.

Reviewer 3 Report

The authors have improved the article according to the comments. The article can now be published.

Author Response

Thank you for your comments.

Round 3

Reviewer 2 Report

The authors have addressed the issues reported in the previous reply. The manuscript can be published in the present form.